# Plurihormonal Pituitary Neuroendocrine Tumors: Clinical Relevance of Immunohistochemical Analysis

**DOI:** 10.3390/diagnostics14020170

**Published:** 2024-01-11

**Authors:** Roxana-Ioana Dumitriu-Stan, Iulia-Florentina Burcea, Ramona Dobre, Valeria Nicoleta Nastase, Raluca Amalia Ceausu, Marius Raica, Catalina Poiana

**Affiliations:** 1Department of Endocrinology, ‘Carol Davila’ University of Medicine and Pharmacy, 020021 Bucharest, Romania; 2‘C. I. Parhon’ National Institute of Endocrinology, 011863 Bucharest, Romania; 3Department of Microscopic Morphology/Histology, ‘Victor Babes’ University of Medicine and Pharmacy, 300041 Timisoara, Romania; 4Angiogenesis Research Centre, ‘Victor Babes’ University of Medicine and Pharmacy, 300041 Timisoara, Romania

**Keywords:** plurihormonal pituitary neuroendocrine tumors, acromegaly, Cushing’s disease, immunohistochemistry, pituitary transcription factors

## Abstract

Plurihormonal pituitary neuroendocrine tumors (PitNETs) are rare forms of tumors that express more than one hormone. The most common association is between growth hormone (GH) and prolactin (PRL), but other unusual combinations have been reported, such as GH and ACTH. Usually, the clinical dominance in these cases is related to GH hypersecretion. In these cases, immunohistochemistry (IHC) of transcription factors (TFs) is very useful for an accurate diagnosis. We included 42 patients diagnosed with pituitary neuroendocrine tumors (PitNETs): 37 patients with a confirmed diagnosis of acromegaly, and 5 patients with prolactinomas. All patients underwent transsphenoidal surgical intervention. We correlated the immunohistochemical features of plurihormonal PitNETs with clinical, hormonal, and imaging data. Tumor specimens were histologically and immunohistochemically examined. Based on the 2022 WHO classification, using IHC, 13 patients exhibited positive staining for more than one hormone, while unusual combinations like GH + ACTH and PRL + ACTH were also identified in other cases. Unusual cell combinations that produce hormones unrelated histogenetically, biochemically, or through regulatory mechanisms can appear and may display aggressive behavior, persistent disease, and high recurrence. We have not identified a clear correlation with the prognosis of these rare PitNETs.

## 1. Introduction

Plurihormonal PitNETs express two or more types of pituitary hormones. To date, we do not have a clear etiology for these tumors, but their pathogenesis is hypothesized to be a result of the neoplastic transformation of two different cell lines or the trans-differentiation of a once-tumor cell line into a different hormone-producing cell line [1].

Clinically, plurihormonal PitNETs are commonly characterized by the hypersecretion of only one type of hormone, though, in some cases, patients are asymptomatic. The most common association reported is between GH and PRL and one or more glycoprotein hormones. The co-secretion of GH and ACTH has also been diagnosed and, in this case, the clinical dominance is related to GH hypersecretion and the presence of subclinical Cushing’s disease. Due to the use of poor diagnostic techniques, plurihormonal PitNETs were often misdiagnosed in the past. Due to the development of technologies such as electron microscopy, immunoelectron microscopy, and immunohistochemistry, and a better understanding of these tumors, the proportion of patients diagnosed with a tumor expressing multiple hormones has increased. At present, most plurihormonal tumors are diagnosed based on the following aspects: clinical symptoms and endocrine activity, imaging and intraoperative findings, histology, immunohistochemistry, and ultrastructure [2].

The hormones identified by IHC do not always have a clinical correspondence as often no elevation of the corresponding serum hormones is detected. A possible reason is that the hormone secreted by the tumor is not biologically active, or it has lost activity after entering the blood circulation [3]. One example is ACTH with a high molecular weight or ACTH hypersecretion in a biologically inactive form that can be present in patients with acromegalic features that are not clinically/biochemically evident (silent corticotroph cells).

The latest World Health Organization (WHO) classification of PitNETs from 2022 is based on transcription factors and differentiation drivers in the differentiation pathway of pituitary cells [4]. The classification also includes a new subtype of PitNET responsible for the secretion of TSH, GH, and PRL: the mature plurihormonal Pit-1 lineage tumor, composed of monomorphous cells. These tumors resemble a mammosomatotroph tumor, secreting GH and PRL, but they also express variable GATA and TSHβ, producing overt acromegaly associated with hyperprolactinemia. The immature subtype of these tumors can cause hyperthyroidism [5].

A better understanding of the etiology of these tumors can help to establish a more accurate diagnosis and to apply a personalized treatment. A very important fact is that the structure of the adult pituitary has a dual embryologic origin: the dorsal and ventral sides regulate the expression of transcription factors through proliferative signals. On the ventral side, activated SF-1 (steroidogenetic factor 1), GATA, and ERα are activated and determine the gonadotroph lineage. A T-Pit signal differentiates dorsal cells into corticotroph cells and Pit-1 induces lactotroph, somatotroph, thyrotroph, and mammosomatotroph cells and plurihormonal lineage tumors [6,7] (Figure 1).

The 2022 WHO classification mentions the existence of high-risk PitNETs (sparsely granulated somatotroph tumors, immature Pit-1 lineage tumors, acidophil stem cell tumors, Crooke cell tumors, ‘silent’ corticotroph tumors, and null cell tumors: see Figure 1). Previous studies have shown that during postnatal life, in the anterior pituitary gland and in pituitary tumors, there is a population of progenitor cells that will differentiate into hormone-producing cell types. These cells may be a possible tumor-initiating population originating from an already-committed cell.

## 2. Materials and Methods

In total, 42 patients diagnosed with pituitary neuroendocrine tumors (PitNETs) were included in our study: 37 patients with a confirmed diagnosis of acromegaly, and 5 patients with prolactinomas. The biochemical diagnosis of acromegaly and prolactinomas was based on The Endocrine Society Clinical Practice Guidelines [8,9].

All patients underwent transsphenoidal surgical intervention. Data regarding clinical, biochemical, and imaging characteristics were collected retrospectively. This retrospective, observational study was conducted following the Declaration of Helsinki and approved by the Institutional Ethics Committee of ‘C. I. Parhon’ National Institute of Endocrinology, Bucharest, Romania (Ethics Approval No. 04/24.02.2022).

The patients underwent pituitary neurosurgical intervention in the Neurosurgery Clinic of ‘Bagdasar Arseni’ Emergency Clinical Hospital (Bucharest, Romania), the Neurosurgery Clinic of ‘Colentina’ Hospital (Bucharest, Romania), the Neurosurgery Clinic of the Brain Institute, Monza Hospital (Bucharest, Romania), or the NeuroHope Clinic (Bucharest, Romania). Using postoperative tumor paraffin blocks, we performed morphological and IHC analysis. Based on the IHC classification, we selected only the plurihormonal PitNETs.

The postoperative tumor blocks underwent morphological and immunohistochemical analysis at the Department of Microscopic Morphology/Histology and Angiogenesis Research Centre, ‘Victor Babes’ University of Medicine and Pharmacy (Timisoara, Romania).

### Histopathological and Immunohistochemical Examination

The histopathological diagnosis was established after routine staining with hematoxylin and eosin (H&E) on 3 µm sections for each case. The quality of the specimens was verified using immunostaining with vimentin (ETU Leica Biosystems Wetzlar, Germany, clone V9, RTU). Morphological staining was performed using a Leica Autostainer XL (Leica Biosystem Newcastle Ltd., Balliol Business Park West, Benton Lane, New Castle Upon Tyne NE 12 EW, UK). Microscopic examination was performed using a Nikon Eclipse E 600 microscope (Nikon Corporation, Tokyo, Japan).

After the histological evaluation of the specimens stained with hematoxylin and eosin, the immunohistochemical hormonal profile was evaluated. The primary antibodies used are given in Table 1.

Bond Epitope Retrieval Solution 1 and 2 with pH values of 6 and 9 were used for unmasking (Leica Biosystems, Newcastle Ltd., Newcastle Upon Tyne NE 12 8EW, UK) and 3% hydrogen peroxide was used to block endogenous peroxidase for 5 min.

The immunohistochemical reactions were assessed at the cellular level. The immuno-histochemical expression of GH, PRL, TSH, ACTH, FSH, and LH was analyzed at the cytoplasmatic level, along with the expression of Ki-67 and Pit-1 in the nucleus. Stains for the 6 pituitary hormones were scored in a blinded fashion. The proportion score for the anterior pituitary hormones was quantified according to the following criteria: score 0 (0–10% positive cells), score 1+ (10–30% positive cells), score 2+ (30–60% positive cells), and score 3+ (>60% positive cells). The intensity scores used were from 0 to 3+ (from absent to strongly stained). A staining superior to 10% was considered positive for the purpose of interpreting the results. The nuclear-positive cells for Ki-67 were quantified using optical microscopy (magnification ×20) using Image J version 2.0 (a semiautomatic evaluation, which excluded endothelial and stromal cells’ nuclei). Pit-1 was only immunohistochemically stained in 2021 and there are results for only three patients.

The short-term outcomes, determined approximately 3 months postoperatively, were as follows: biochemical remission in the absence of adjuvant medical treatment, and evaluation of the residual tumor using magnetic resonance imaging (MRI) or computed tomography (CT). The patients had an average follow-up period of 5 years.

## 3. Results

Out of the 42 patients included in the study, only 13 patients with plurihormonal PitNETs were identified. We established the diagnosis of plurihormonal PitNET in all tumors that had positive IHC expression for two or more hormones. A notable exception was the synchronous co-expression of GH and PRL: their co-expression without an additional hormone indicates a mammosomatotroph adenoma and does not meet the criteria for a plurihormonal tumor.

Our study included eight females (61.5%) and five males (38.46%), with a mean age of 47.15 years (age range: 33–69).

Based on the 2022 WHO classification, we grouped the cases according to their IHC characteristics (Table 2).

In the cases of the plurihormonal PitNETs identified, only one case had a Ki-67 of 3.2% (GH + PRL + TSH + LH). On histopathological examination, nine cases were acidophils, one was mixed, and three cases were cromophobes. The main architectural pattern was papillary.

The positivity rate was highest for GH immunohistochemistry and was correlated with high serum hormone levels in all cases. ACTH immunohistochemistry was positive in two patients, but only one had endocrine symptoms.

In one case, all pituitary hormones had positive immunohistochemistry. The patient had endocrine symptoms related to GH hypersecretion.

### 3.1. Clinical Features

The majority of the patients had clinical features related to GH hypersecretion (11 patients). One case was diagnosed with prolactinoma and one case had clinical features specific to a mixed GH- and PRL-secreting PitNET. An unusual histological combination was the one between GH and ACTH, with only a few cases having been reported to date in the literature. One patient, who was diagnosed with acromegaly with positive immunohistochemical staining for GH, ACTH, PRL, and LH, showed high ACTH (normal laboratory range: 3–66 pg/mL) and cortisol levels (normal laboratory range: 4.82–19.5 ug/dL), with a lack of suppression in dexamethasone tests.

High IGF1 and GH (nadir in OGTT—oral glucose tolerance test) levels at diagnosis were detected in 11 cases, with a mean value of 3.47 ± 1.56 (×ULN—upper normal limit), where 3 cases had hyperprolactinemia and 2 cases presented with mixed PRL- and GH-secretion. The patient with PRL-secreting PitNET had pituitary apoplexy at onset and underwent transsphenoidal intervention.

There were no cases with a high TSH level or elevated FSH/LH. High ACTH levels (>20 pg/mL) were identified in two cases. One case had clinical Cushing’s features (red face, muscle weakness, depression, and insomnia).

Among the 13 plurihormonal PitNETs, 10 cases had concordant clinical and histological diagnoses (Table 3).

The main clinical symptoms at onset included headache and vision defects, irregular menses, and the acromegalic phenotype of enlarged extremities and a prominent jaw.

Pit-1 was only immunohistochemically stained in 2021 and there are results for transcription factors from only three patients (Table 4). Intense Pit-1 positivity was observed in all cases.

### 3.2. Radiological Evaluation

Based on radiological assessment, macroadenomas were diagnosed in 12 cases and there was 1 case of a microadenoma. The mean preoperative maximal tumor diameter at diagnosis was 20.7 ± 7.6 mm.

Invasion of the cavernous sinus was present in four cases: one case was Knosp grade 1, two cases were Knosp grade 2, and one case was Knosp grade 3 (Figure 2).

All patients underwent transsphenoidal intervention as first-line treatment. Only five patients underwent medical therapy (with cabergoline) preoperatively.

After transsphenoidal surgery, one case was cured and the other cases achieved biochemical control under medical treatment. The surgical outcome was that in two cases, gross total resection (GTR) was possible.

### 3.3. Clinical Cases

Among the patients included in our study, we identified several cases with unusual immunohistochemical combinations, which were associated in the literature with a negative prognosis (Figure 3 and Figure 4).

## 4. Discussion

The most common association identified in the cases of plurihormonal PitNETs is between GH and PRL, but other, quite rare associations, like GH and ACTH, were reported in 21 cases in the literature [8]. In these cases, the clinical dominance is related to GH hypersecretion with subclinical Cushing’s disease. Regarding these cases, one hypothesis might be the presence of high-molecular-weight ACTH, which is in a biologically inactive form in patients with acromegalic features, or ACTH hypersecretion that is not clinically/biochemically evident (silent corticotroph cells) [8,9].

The pathogenesis of plurihormonal tumors may be a result of the neoplastic transformation of two different cell lines or the trans-differentiation of a once-tumor cell line into a different hormone-producing cell line. We do not have a clear correlation with the prognosis. In such cases, the use of IHC for pituitary TFs is very useful. Unusual cell combinations that produce hormones unrelated histogenetically, biochemically, or through regulatory mechanisms can appear and may display aggressive behavior, persistent disease, and high recurrence. Considering that the IHC findings for pituitary hormones are often focal, very weak, or uncertain, TF staining may serve as a critical determinant for histological diagnoses in such instances.

Based on the 2022 WHO classification, a pathological examination protocol must include immunostains of the transcriptions factors (Pit-1, T-Pit, and SF-1) for the determination of cell lineage, and then the following stains for possibly associated hormones based on the results of the TF stains: GH, PRL, and TSH for PIT-1-positive adenomas, only ACTH for T-Pit-positive adenomas, and LH and FSH for SF-1-positive adenomas. Several TFs have been found to regulate cellular differentiation of the adenohypophysis, and they are also essential for the differentiation and maturation of the neuroendocrine cells from Rathke’s pouch [10].

Challenges can arise in cases with discrepancies between the clinical manifestations and the histology. A gonadotroph adenoma positive for SF-1, and lacking FSH and LH, was the most frequently found discrepancy. The second most common tumor type was the null cell adenoma with positive staining for ACTH [10].

In other earlier studies, when TF stains were not available, 66.3% of clinically endocrine-inactive pituitary adenomas were negative for any pituitary hormone stain and were classified as null cell adenomas [11].

Some types of plurihormonal PitNETs are difficult to diagnose. Several studies confirmed that the hormones secreted by pituitary adenomas in vitro were different from those observed using immunohistochemistry and serology. In these cases, hormonal examination using a tumor cell supernatant cultured in vitro is a good diagnostic method [12]. Using this method, it has been shown that non-functional PitNETs can secrete LH, FSH, and PRL in cell cultures in vitro [13,14]. The immunohistochemistry result can contrast with in vitro culture results if the cell hormone level is low, or if tumor cells synthesize hormones but are not secreted outside the cell or they degrade immediately after secretion. Another possibility is that the hormones have only immunological, and not biological, activity [15,16,17].

In the literature, the Pit-1-positive Pit-NETs with positive stains for multiple Pit-1 lineage hormones are associated with aggressive behavior [18,19]. In our study, we identified one case of Pit-1-positive plurihormonal PitNET that showed aggressive behavior: a resistance to medical treatment after transsphenoidal surgery. The other two cases with positive staining for Pit-1 showed GH and LH expression on IHC evaluation and were controlled under medical treatment after transsphenoidal intervention.

The Ki-67 labeling index had, in the majority of cases, a value below 3%. Only one female patient with IHC positivity for GH, PRL, TSH, and LH had a Ki-67 of 3.2% (we did not have a result for Pit-1 available in this case) and the clinical manifestations were specific for a mixed GH- and PRL-secreting PitNET.

To date, only one large study, which included 70 plurihormonal PitNETs, has shown that, in addition to IHC, the examination of the hormones in tumor culture supernatant is helpful for diagnosis. The SOX2 stem cell markers, stained in 12 patients diagnosed with plurihormonal tumors, had a positivity rate of 33%, but the pathogenesis of these tumors cannot be explained completely by stem cell origin theory [20,21,22,23].

Histologically, plurihormonal PitNETs represent 10–15% of all pituitary tumors and up to 1% in unselected autopsy studies [24,25,26,27]. Another important fact is that, even though rare, there is a possibility of two or more distinct adenomas coexisting in the same pituitary gland. Double pituitary adenomas are most frequently GH- or ACTH-secreting, with IHC stains positive for GH and ACTH, that can cause Cushing’s disease or acromegaly [28,29,30]. In our case study, we did not identify multiple PitNETs or synchronous PitNETs.

The WHO classification standardizes diagnosis for patient care and guides general pathologists alongside the five-tier classification that combines pathological features (cell differentiation and proliferative markers) with radiological parameters (invasion), which can be used for a better diagnosis and risk stratification [31,32,33,34,35,36]. Although these classification systems can be useful for tumor management, tumor staging may not predict tumor behavior, as invasiveness does not necessarily imply aggressive behavior.

In our study, only two cases showed aggressive behavior: one Pit-1-positive case with IHC stains positive for GH and ACTH, and one case with IHC stains positive for PRL and ACTH (a PRL-secreting PitNET).

The diagnosis of plurihormonal PitNETs, in our study, was based on clinical symptoms, serum hormone levels, and IHC stains. In 10 cases, we observed concordance between the IHC stains and the hormonal levels.

The diagnosis of these tumors is controversial, and IHC, hormone levels, and clinical symptoms may not be sufficient. Another useful tool for diagnosis might be genetic testing for mutations, including HMGA2, BMI1, AIP (aryl hydrocarbon receptor-interacting protein), RAS family proteins, and PTEN [37,38,39,40].

The co-staining of multiple hormones using fluorescent antibodies, along with immunohistochemical evaluation, is another approach that can establish a more accurate diagnosis. Currently, there is a limited understanding of pituitary autoimmunity. Also, the data in the literature show a large variability in the use of this method and, to date, the 2022 WHO classification or the five-tiered clinico-pathological classification of PitNETs does not include the detection of fluorescent antibodies as a step in the classification of these tumors [41].

Other techniques that could help clinicians identify and offer a personalized treatment for patients with plurihormonal PitNETs are single-cell sequencing technologies. These have emerged as powerful tools for comprehensively understanding genetic and functional heterogeneity at single-cell resolution. The method can compare tumor and normal pituitary tissue, but this can be challenging due to the mixed pituitary cells that can exist in the pituitary. The limited data, obtained using these methods, in the literature showed that gonadotroph tumors predominantly exhibited downregulated genes, while somatotroph and lactotroph tumors mainly exhibited upregulated genes [42]. Using these methods, the lack of intra-tumoral heterogeneity in several multiple-hormone tumors has been demonstrated as the transcription factors or hormone genes of different lineages were clearly co-expressed in individual cells [42].

## 5. Conclusions

Plurihormonal PitNETs represent a more aggressive type of tumor, which can require a multimodal treatment as often gross total resection is not achieved, and, in the majority of cases, is hard to diagnose. In our study, in only one case was the patient cured by gross total resection. The other patients needed medical treatment after the surgical intervention. One single case had clinical features of acromegaly, hyperprolactinemia, and Cushing’s disease, which was concordant with the IHC staining.

Unfortunately, only three patients were tested for Pit-1 positivity and transcription factor evaluation must be mandatory for a more accurate diagnosis and risk stratification.

Our study brings new data in addition to the plurihormonal PitNET cases already reported in the literature. Larger studies are needed to better understand the etiology and to identify the prognosis factors in these cases. Novel techniques, like single-cell sequencing technologies, could be used in the future for a better understanding of these tumors.

## Figures and Tables

**Figure 1 diagnostics-14-00170-f001:**
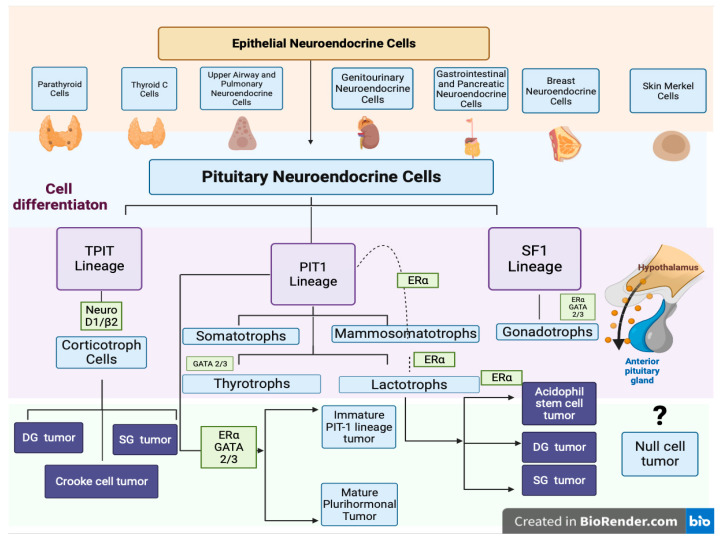
Pituitary cell differentiation and PitNETs, adapted from the 2022 WHO classification of PitNETs (created with Biorender, Toronto, ON, Canada).

**Figure 2 diagnostics-14-00170-f002:**
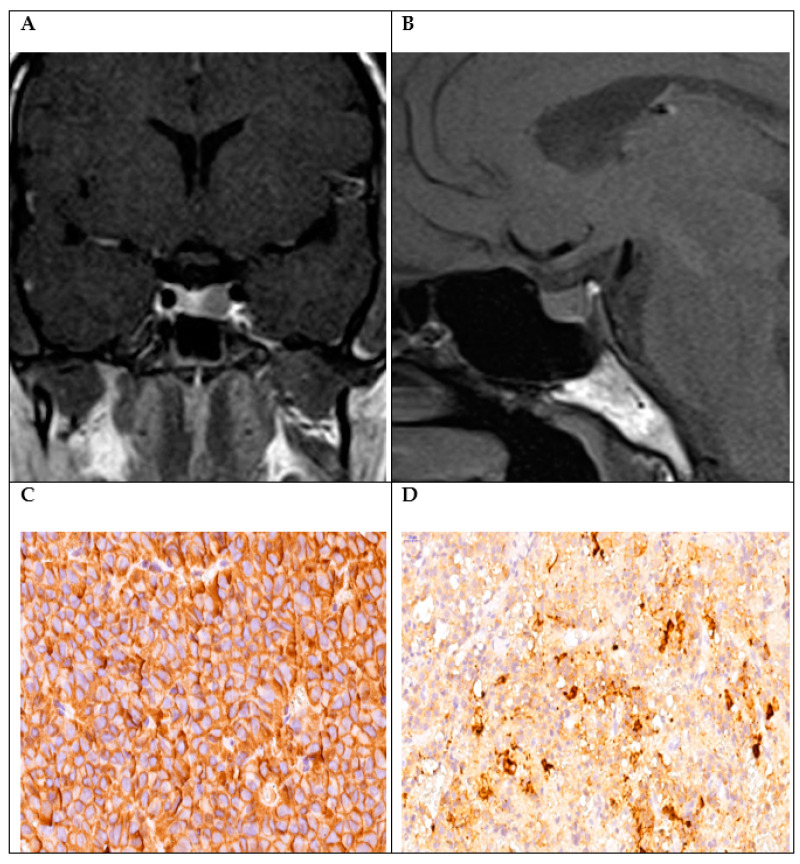
CT (computed tomography) scans of Pit-1-positive plurihormonal PitNET. Preoperative CT scan, coronal section: pituitary microadenoma, 8.6/3.9 mm (**A**); postoperative CT scan, sagittal section: residual tumor of 7/4 mm (**B**); IHC positivity for GH (+3, ×60 magnification) (**C**); and ACTH (+1, 1, ×40 magnification) (**D**).

**Figure 3 diagnostics-14-00170-f003:**
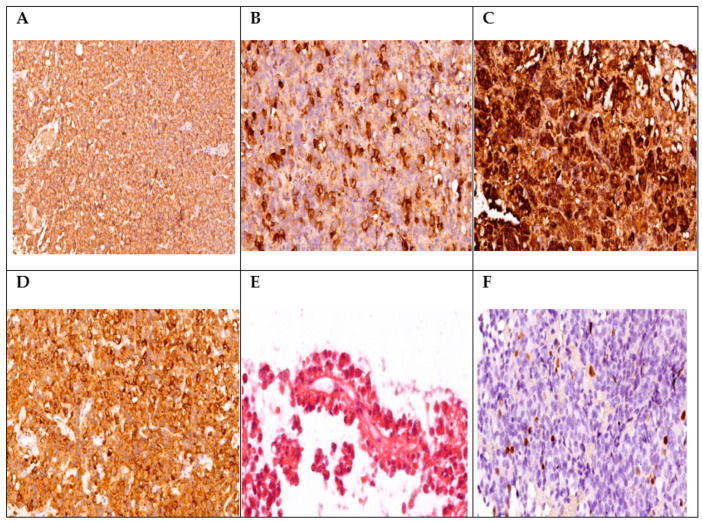
A case of acromegaly with IHC positivity for GH, PRL, ACTH, and LH with development of Cushing’s disease-specific symptoms. IHC staining for GH (+3, ×20 magnification) (**A**); PRL (+3, ×20 magnification) (**B**); ACTH (+1, ×25 magnification), (**C**); and LH (+1, ×30 magnification) (**D**). Hematoxylin and eosin (H&E) staining (×60 magnification: acidophilic pituitary adenoma, solid, and a papillary component) (**E**); Ki-67 < 3% (**F**).

**Figure 4 diagnostics-14-00170-f004:**
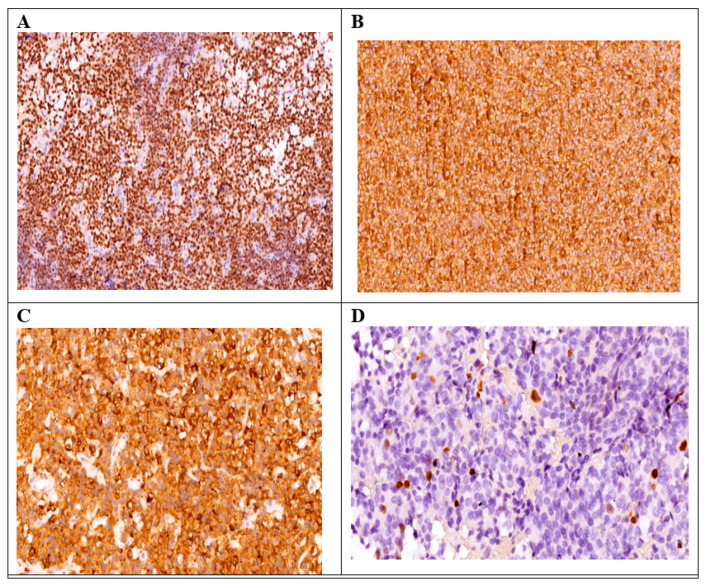
Male patient diagnosed with acromegaly. IHC staining for Pit-1 (+3, ×20 magnification) (**A**); GH (+3, ×30 magnification) (**B**); LH (+3, ×30 magnification), (**C**); and Ki-67 < 3% (**D**).

**Table 1 diagnostics-14-00170-t001:** Antibodies used for IHC of transcription factors and pituitary hormones.

	Company	Clone	Dilution Factor	Expression Pattern
GH	Dako, Agilent	Polyclonal rabbit anti-human	1:400	Cytoplasmatic
PRL	Dako, Agilent	Polyclonal rabbit anti-human	1:300	Cytoplasmatic
ACTH	Dako, Agilent	C93	1:50	Cytoplasmatic
FSH	Thermo Fisher Scientific	FSH03	1:500	Cytoplasmatic
LH	Thermo Fisher Scientific	LH01	1:500	Cytoplasmatic
TSH	Thermo Fisher Scientific	TSH01 + TSH02	1:400	Cytoplasmatic
Ki-67	Thermo Fisher Scientific	MM1, RTU	-	Nuclear
Cytokeratin Cam 5.2	Diagnostic BioSystems	CAM5.2, RTU	-	Cytoplasmatic
Pit-1	Thermo Fisher Scientific	Rabbit polyclonal antibody	1:500	Nuclear

GH—growth hormone; PRL—prolactin; ACTH—adrenocorticotropic hormone; TSH—thyroid-stimulating hormone; FSH—follicle-stimulating hormone; LH—luteinizing hormone; Pit-1—pituitary-specific transcription factor Pit-1; Dako, Agilent, Santa Clara, CA, USA; Thermo Fisher Scientific, Waltham, MA, USA; Diagnostic Bio-Systems, Pleasanton, CA, USA.

**Table 2 diagnostics-14-00170-t002:** Immunohistochemical classification.

IHC	*n* (%)	Histological Features
GH + ACTH	1 (2.38%)	Diffuse strong staining for GH (+3) Minority ACTH cells (+1)
GH + LH	2 (4.76%)	Diffuse strong staining for GH (+3) Minority LH cells (+1)
GH + TSH + LH	2 (4.76%)	Distinct groups of GH, TSH, and LH cells
GH + PRL + FSH	2 (4.76%)	Diffuse strong staining for GH and PRL (+3) Minority FSH cells (+1)
GH + PRL + TSH + FSH/LH	4 (9.52%)	Distinct groups of GH, PRL, TSH, and FSH/LH Diffuse strong staining for GH
GH + PRL + ACTH + LH	1 (2.38%)	Distinct groups of ACTH, GH, PRL, and LH cells Diffuse strong staining for GH (+3)
PRL + ACTH	1 (2.38%)	Strong PRL staining (+3) Minority ACTH cells (+1)

GH—growth hormone; PRL—prolactin; ACTH—adrenocorticotropic hormone; TSH—thyroid-stimulating hormone; FSH—follicle-stimulating hormone; LH—luteinizing hormone.

**Table 3 diagnostics-14-00170-t003:** Clinicopathologic features of patients with plurihormonal PitNETs.

Number		Age at Diagnosis, Gender	Clinical Features	Radiologic Features	Elevated Hormone Levels	IHC Positivity	Knosp Grade
1	GH + PRL + TSH + LH	69, Female	Acromegaly Hyperprolactinemia	Macroadenoma	IGF1, PRL	GH (+3) PRL (+2) TSH (+2) LH (+1)	0
2	GH + PRL + ACTH + LH	44, Female	Acromegaly Hyperprolactinemia Subclinical Cushing’s	Microadenoma	IGF, GH, PRL, ACTH, and cortisol	GH PRL ACTH LH	0
3	GH + PRL + TSH + LH	55, Female	Acromegaly	Macroadenoma	IGF1, GH, PRL	GH (+3) PRL (+2) TSH (+2) LH (+1)	1
4	GH + PRL + FSH	33, Female	Acromegaly	Macroadenoma	IGF1, GH	GH (+3) PRL (+1) FSH (+1)	2
5	GH + PRL + LH	55, Female	Acromegaly	Macroadenoma	IGF1, GH	GH (+3) PRL (+1)	0
6	GH + ACTH	50, Female	Acromegaly	Macroadenoma Cavernous sinus invasion	IGF1	GH (+3) ACTH (+1)	1
7	GH + PRL + TSH + FSH	49, Male	Acromegaly	Macroadenoma Cavernous sinus invasion	IGF1, ACTH	GH (+3) PRL (+3) TSH (+1) FSH (+2)	2
8	GH + LH	32, Male	Acromegaly	Macroadenoma	IGF1, GH	GH (+3) LH (+3)	0
9	GH + LH	43, Male	Acromegaly	Macroadenoma	IGF1, GH	GH (+3) LH (+2)	0
10	GH + TSH + LH	47, Male	Acromegaly	Macroadenoma	IGF1, GH	GH (+3) TSH (+3) LH (+3)	0
11	GH + PRL + FSH	36, Female	Acromegaly	Macroadenoma Cavernous sinus invasion	IGF1, PRL	GH (+2) PRL (+1) FSH (+1)	2
12	GH + PRL + TSH + LH	63, Female	Acromegaly, Hyperprolactinemia	Macroadenoma	IGF1, PRL	GH (+3) PRL (+1) TSH (+2) LH (+1)	0
13	PRL + ACTH	37, Male	Hyperprolactinemia	Macroadenoma Cavernous sinus invasion	PRL, ACTH	PRL (+3) ACTH (+1)	3

GH—growth hormone; PRL—prolactin; ACTH—adrenocorticotropic hormone; TSH—thyroid-stimulating hormone; FSH—follicle-stimulating hormone; LH—luteinizing hormone.

**Table 4 diagnostics-14-00170-t004:** Transcription factors correlated with IHC diagnosis and clinical data.

Case Number	IHC	Pit-1	Tpit	SF-1	Ki-67	CAM 5.2 Expression	Clinical Manifestations	Resistance to Treatment
1	GH + ACTH	+1	0	0	<3%	No expression	Acromegaly	Yes
2	GH + LH	+3	0	+1	<3%	Perinuclear (densely granulated)	Acromegaly	No
3	GH + LH	+3	0	+3	<3%	Perinuclear (densely granulated)	Acromegaly	No

GH—growth hormone; PRL—prolactin; ACTH—adrenocorticotropic hormone; LH—luteinizing hormone; the IHC results for transcription factors were quantified using a positivity score.

## Data Availability

The original contributions presented in the study are included in the article/supplementary material, further inquiries can be directed to the corresponding author/s.

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
