# Peer review of "Plurihormonal Pituitary Neuroendocrine Tumors: Clinical Relevance of Immunohistochemical Analysis"

_diagnostics, 2024, doi:10.3390/diagnostics14020170_

Round 1

Reviewer 1 Report

Comments and Suggestions for Authors

Dumitriu-Stan and colleagues present clinical cases of plurihormonal expression in pituitary adenomas. They analyzed 42 cases including 13 that demonstrated plurihormonal expression. They detected each hormone by individual immunostaining and set up a scaling system to compare staining across patients, counting cells positive for each hormone. They present their finding and the diversity of hormonal expression in pituitary adenomas with the different combination in a small set of samples.

This study is interesting and needed in the field. As the authors are explaining, often this tumor type can be ignored if not all hormones are experimentally assessed during pathology. The work is well-described with several tables characterizing all samples. 

Comments for the authors:

1- The term PitNETs is not accepted in the field, and it would be best to use pituitary adenomas, or at least pituitary adenomas/PitNETs. 

2- Several sentences, specifically in the introduction would require editing (see section about English language).

3- The authors may have included co-staining using fluorescent antibodies to  make they case stronger. I understand that in regular pathology, each hormone may be detected individually, but in a scientific report it would be interesting to see in the same section co-staining of multiple hormones for a better assessment. Tumor heterogeneity is important, and different sections of the adenoma may present with different cells. Having the same sections stained with all antibodies would be preferable. 

4- Minor comment: line 60, the author may like to change "BTSH" to "TSHB". 

5- More up to date techniques, including single-cell transcriptomics would bring even more to the characterization of these samples. 

Comments on the Quality of English Language

Several sentences need editing.

A few examples:

- lines 37-38 is not understandable as is: "Clinically, plurihormonal PitNETs are commonly characterized by the expression of one type of hormone, or patients are asymptomic." It seems something is missing, unclear what they authors want to convey here. 

- Lines 43-44: "With the development [..], the proportion of patients diagnosed now has been increasing." Again something seems missing and I would add the proportion of patients diagnosed with a tumor expressing multiple hormones has been increasing. 

- Lines 49-50 "The hormones identified by IHC analysis do not always have clinical correspondence with no elevation of the corresponding serum hormones." I would suggest to change to The hormones identified by IHC do not always have clinical correspondence as often no elevation of the corresponding serum hormones are detected. 

- Line 142 "does not meets" remove the s at "meets"

Author Response

Thank you for all the comments and suggestions. I have rechecked, point by point, all the suggestions and I made changes to the manuscript.

Reviewer 2 Report

Comments and Suggestions for Authors

A supplement [if possible] with some more details of the patients' clinical presentation and laboratory evaluation could be handy, the stuff on pages 5/6 are still sketchy. For example what do "high ACTH and cortisol" stand for? Also correct:  there is no "lack of inhibition" ln 165 but of suppression.

The authors could stress the novelty of their work/findings and expand on any clinical usefulness

Otherwise OK

Comments on the Quality of English Language

More or less Ok

Author Response

(The authors gave the same response as above.)
